# High-Definition Trans-Spinal Current Stimulation Improves Balance and Somatosensory Control: A Randomised, Placebo-Controlled Trial

**DOI:** 10.3390/biomedicines12102379

**Published:** 2024-10-18

**Authors:** Teni Steingräber, Leon von Grönheim, Michel Klemm, Jan Straub, Lea Sasse, Jitka Veldema

**Affiliations:** Department of Sport Science, Faculty of Psychology and Sports Science, Bielefeld University, 33501 Bielefeld, Germany; teni.steingraeber@uni-bielefeld.de (T.S.);

**Keywords:** high-definition direct current stimulation, high-definition alternating current stimulation, spinal cord, balance, lower limbs, deep sensitivity, superficial sensitivity

## Abstract

Objectives: To investigate and compare the effects of three different high-definition (HD) non-invasive current stimulation (NICS) protocols on the spinal cord on support balance and somatosensory abilities in healthy young people. Methods: Fifty-eight students were enrolled in this crossover study. All participants underwent application of (i) 1.5 mA anodal high-definition trans spinal direct current stimulation (HD-tsDCS), (ii) 1.5 mA cathodal HD-tsDCS, (iii) 1.5 mA high-definition trans spinal alternating current stimulation (HD-tsACS), and (iv) sham HD-tsDCS/ACS over the eighth thoracic vertebra in a randomised order. Balance (Y Balance test), deep sensitivity (Tuning Fork Test), and superficial sensitivity (Monofilament Test) of the lower limbs were tested immediately before and after each intervention. Results: Balance ability improved significantly following anodal HD-tsDCS and HD-tsACS compared with that following sham HD-tsDCS/ACS. Similarly, deep sensitivity increased significantly with anodal HD-tsDCS and HD-tsACS compared to that with sham HD-tsDCS/ACS and cathodal HD-tsDCS. Furthermore, superficial sensitivity improved significantly following anodal HD-tsDCS compared with that after HD-tsACS and cathodal HD-tsDCS. Conclusions: Our data show that HD-tsNICS effectively modulates the balance and somatosensory control of the lower limbs. Several diseases are associated with illness-induced changes in the spinal network in parallel with sensorimotor disabilities. Non-invasive spinal modulation may be a favourable alternative to conventional brain applications in rehabilitation. Future studies should therefore investigate these promising approaches among cohorts of patients with disabilities.

## 1. Introduction

Non-invasive current stimulation (NICS) is a powerful tool that involves the modulation of neural processing, the use of which continues to grow in research and therapy. In the present study, we investigated the utility of this method in the field of sensorimotor control, with a particular focus on posture and balance.

### 1.1. The Spinal Cord as a Promising Alternative to the Brain for Application of NICS in Supporting Balance Control

Existing data have shown that balance and posture are highly complex functions controlled by extensive brain and spinal networks that continuously interact with each other [1,2,3,4]. Numerous studies using (functional) magnetic resonance imaging ((f) MRI) and positron emission tomography (PET) have demonstrated that the brainstem, cerebellum, basal ganglia, thalamus, hippocampus, inferior parietal cortex, and frontal lobe all play key roles in this process [1,2]. Investigations of the Hoffmann reflex (H-reflex) have indicated that the spinal cord is crucially involved in balance and postural control [3,4,5,6,7,8]. In fact, numerous studies have demonstrated that both balance training and improved balance control are accompanied by suppressed H-reflex. In contrast, disability-induced worsening of balance control correlates with H-reflex increase. For example, in addition to an improved balance ability, a single training session on a balance platform induced a short-term suppression of the H-reflex in healthy elderly subjects [5]. Similarly, as compared to normal walking, balancing on a narrow beam induced a short-term reduction of the H-reflex amplitude in healthy people [6]. In parallel with gait and balance disability, stroke patients demonstrated a long-term increase in H-Reflex amplitude compared to healthy peers [7]. The successful balance rehabilitation in a stroke cohort was associated with H-reflex normalisation [8]. It is cogitable that the reactions of the spinal reflex system during postural disturbances are not always adequate. The reflex-mediated excessive stabilising movements often paradoxically lead to destabilisation. Suppression of the spinal reflexes may increase the involvement of supraspinal centres during motor control and prevent reflex-mediated destabilising movements [3]. A model first described in 1911 [9] suggested that central pattern generators (CPGs) play a crucial role in balance and posture [10,11,12]. These self-organising neural circuits, located within the lumbar and cervical spinal cord intumescences, are responsible for the coactivation of both agonist and antagonist muscles during rhythmic and stereotyped motor actions such as walking or swimming [10,11,12]. Several studies have further pointed out that CPGs not only act as “tact generators” of several motor actions, but also as “dampeners” of disturbing reflexes (e.g., evoked by stumbling) through the facilitation or inhibition of alpha motoneurons [10]. It is also assumed that self-regulating CPG circuits work under the control of the brainstem, without input from higher brain regions [10,11,12].

Existing data [3,4,8,9,10,11,12] strongly indicate that the spinal cord is a promising target for non-invasive neuromodulation to support balance abilities. Despite this, existing research has thus far predominantly focused on the primary motor cortex (M1) and the cerebellum [13,14,15]. However, our earlier study encouraged the investigation of spinal cord regions [16]. In this study, the direct comparison of NICS over the (i) M1, (ii) cerebellum, and (iii) spinal cord demonstrated that spinal application induced superior improvement in balance ability compared to “conventional” M1 and cerebellar stimulation [16]. The present study examined spinal NICS more closely than previous studies.

### 1.2. Stimulation Parameters for Spinal NICS in Supporting Balance Control in Healthy Individuals

NICS involves the application of a low-intensity current that flows between the anode and cathode. Several factors, including stimulation duration, electrode size and positioning, and current direction and intensity, can significantly determine NICS-induced neurophysiological effects [17,18,19]. Essentially, we distinguish between (i) anodal direct current stimulation (DCS), with the anode placed over the region of interest and the cathode placed over another region; (ii) cathodal DCS with reverse electrode positioning; and (iii) alternating current stimulation (ACS), comprising a current that changes direction [20,21,22]. Currently, anodal DCS is the preferred protocol for experiments in healthy individuals, compared with either cathodal DCS or ACS [23,24]; this also applies to balance and postural control [25,26,27]. The reason for this phenomenon is a generally known theory that anodal DCS “supports” while cathodal DCS “suppresses” neural networks and abilities. However, numerous studies have demonstrated NICS-induced modulation outside of this simplified framework [28,29], indicating that both cathodal DCS and ACS should be investigated more intensively. Therefore, our study aims to directly compare the effects of three protocols.

Further criteria that significantly determine NICS-induced effects include the electrode size and positioning. “Conventional” NICS uses two rectangular electrodes with dimensions of 70 mm × 50 mm. Although various electrode shapes and sizes are available and have been approved for human use [30], their applications in previous studies remain underrepresented. Our study investigated high-definition (HD) NICS. During HD-NICS, the current is delivered through one small round electrode (placed over the region of interest), surrounded by multiple small electrodes (or alternatively, one large ring electrode) [31,32,33]. Computer models have repeatedly demonstrated that HD-NICS modulates the neural networks in a more targeted manner than “conventional” electrodes [31,32,33]. Accordingly, several original studies have detected differential neural and/or behavioural effects of both methods [31,32,33]. However, no clear statement has yet been made regarding the superior efficacy of one of these approaches.

### 1.3. Investigation of Somatosensation in Parallel to Balance

Somatosensation, defined as the ability to identify and interpret information regarding the state of the body and its physical interactions with the environment, is essential for effective motor control [34,35]. Somatosensory deficits can be found in several patient groups (e.g., stroke, spinal cord injury, Parkinson’s disease, multiple sclerosis, and Guillain–Barre syndrome) in parallel with motor disabilities [36,37,38,39,40], and are associated with worse health status, lower quality of life, and increased risk of injuries and falls [36,37,38,39,40]. Somatosensory impairment is one of the greatest obstacles in successful motor rehabilitation. Limited somatosensory abilities have been shown to be correlated with poorer balance abilities in both healthy elderly and disabled cohorts [38,39,40,41]. However, investigations into the methods to support somatosensory control have often been overshadowed by motor research. In this study, we decided to shed more light on this relevant area and investigate both the deep and superficial somatosensitivity of the lower limbs in parallel with balance control.

## 2. Methods

### 2.1. Study Design

This randomised, placebo-controlled crossover study involved the application of single interventional sessions of (i) anodal HD-tsDCS, (ii) cathodal HD-tsDCS, (iii) HD-tsACS, and (iv) sham HD-tsDCS/ACS in a randomised (PC-generated) order, with a washout period of at least 48 h in between each session. Balance ability and superficial and deep sensitivity were tested immediately before and after each intervention. This trial was performed in accordance with the standards established by the Declaration of Helsinki and approved by the Ethics Committee of Bielefeld University (EUB-2023-080).

### 2.2. Participants

Individuals matching the following criteria were included: (1) age between 18 and 30 years; (2) no contraindications for non-invasive neuromodulation [42]; and (3) no relevant neurological, psychiatric, or orthopaedic disorders. All participants provided written informed consent before participation. Sample size calculation using G*power (version 3.1.9.7) analysis (effect size = 0.25, α error probability *p* < 0.05, Power = 0.95) indicated that a minimum of 40 subjects was required to detect statistically significant effects using ANOVA with four interventions and two time points.

### 2.3. Intervention

The interventions were applied using DC-stimulator PLUS (NeuroConn Gmbh, Ilmenau, Germany) with one small round electrode (3 cm^2^) and one large ring electrode (31 cm^2^). A small round electrode was placed centrally over Th8 using the palpation method [43,44], while a large ring electrode was placed around the central electrode. Four separate 20-min sessions of (a) anodal 1.5 mA HD-tsDCS (anode central), (b) cathodal 1.5 mA HD-tsDCS (cathode central), (c) 1.5 mA HD-tsACS (at a frequency of 10 Hz), and (d) sham HD-tsDCS/ACS (stimulator turned off after 5 s) were applied to each proband (Figure 1).

### 2.4. Assessments

Three different assessments were used to evaluate balance (Y Balance test) and sensitivity (tuning fork and monofilament tests) (Figure 2). The right and left legs were further tested in a randomised order during each assessment. Severe and less severe intervention-induced adverse effects have also been documented. The evaluators and the participants were blinded to the intervention allocation.

During the Y Balance Test [45], the participants performed a maximal reach of the free lower leg in the (a) anterior, (b) posterolateral, and (c) posteromedial directions during a one-leg stance on the opposite leg using a test kit (FMS, Chatham, MA, USA) (Figure 1(Ba). Five trials were conducted for each leg and direction. The mean values of five trials were used for the analysis. A greater reach distance indicated better balance ability.

During the Tuning Fork Test [46,47], the vibration sensitivities of the (a) first metatarsal-phalangeal joint, (b) malleolus medialis, (c) malleolus lateralis, (d) patella, and (e) anterior superior iliac spine were tested (Figure 1(Bb)). A Rydel–Seiffer tuning fork (Kirchner and Wilhelm GmbH + Co. KG, Asperg, Germany) with a scale of 0 to 8 was used. Three trials were conducted at each location. The mean values of three trials were used in the analysis. A higher score indicated better vibration sensitivity.

During the Monofilament Test [48], the pressure sensitivities of the feet, upper and lower legs, and torso were investigated at 27 separate locations (Figure 1(Bc)). Twenty Semmes Weinstein Monofilaments with different thicknesses (1.65, 2.36, 2.44, 2.83, 3.22, 3.61, 3.84, 4.08, 4.17, 4.31, 4.56, 4.74, 4.93, 5.07, 5.18, 5.46, 5.88, 6.10, 6.45, 6.65) were used (Fabrication Enterprises Inc., New York, NY, USA). Each location was touched three times consecutively with each monofilament. Two or more correctly perceived touches were considered as “felt”. A lower score indicated better superficial sensitivity.

### 2.5. Analysis

SPSS software package version 27 (International Business Machines Corporation Systems, Armonk, NY, USA) was used to analyse data collected during the study. Repeated measure ANOVAs with the factors “intervention” and “time” were applied to compare the pre–post changes across interventions. Mauchly’s sphericity tests and Greenhouse–Geisser corrections were applied. Pearson’s correlation was further used to evaluate the relationship between the intervention-induced changes in balance and somatosensation. Pre-interventional comparability was checked using independent sample *t*-tests. A *p*-value of ≤0.05 was considered statistically significant.

## 3. Results

In total, 58 students (age 24.4 ± 2.2 years, 28 females, 30 males, 47 right-footed, 11 left-footed) were tested. The foot each of the individuals favoured when kicking a ball was considered dominant [49]. The interventions were well tolerated without any severe or less severe adverse events. Table 1 lists the data collected during the experiments. Figure 3 presents the intervention-induced changes.

### 3.1. Y Balance Test

The total balance ability score improved significantly with both anodal HD-tsDCS (F1,57 = 11.997; *p* = 0.001) and HD-tsACS (F1,57 = 4.430; *p* = 0.040) compared to sham HD-tsDCS/ACS. Both the right (F1,57 = 5.952; *p* = 0.018) and left (F1,57 = 11.400; *p* = 0.001) legs showed significant improvements after anodal HD-tsDCS compared with sham HD-tsDCS/ACS.

### 3.2. Tuning Fork Test

The deep sensitivity total scores improved significantly after HD-tsACS (F1,57 = 7.501; *p* = 0.008; F1,57 = 11.844; *p* = 0.001) and anodal HD-tsDCS (F1,57 = 4.627; *p* = 0.036; F1,57 = 5.294; *p* = 0.025) compared to sham HD-tsDCS/ACS and cathodal HD-tsDCS, respectively. The right leg improved significantly after HD-tsACS (F1,57 = 5.576; *p* = 0.022) compared with cathodal HD-tsDCS, while the left leg improved significantly after HD-tsACS (F1,57 = 6.525; *p* = 0.013; F1,57 = 6.130; *p* = 0.016) compared to sham HD-tsDCS/ACS and cathodal HD-tsDCS, respectively, and (ii) after anodal HD-tsDCS (F1,57 = 4.013; *p* = 0.050) compared to sham HD-tsDCS/ACS.

### 3.3. Monofilament Test

The total superficial sensitivity score improved following anodal HD-tsDCS (F1,57 = 7.127; *p* = 0.010; F1,57 = 4.287; *p* = 0.043) compared to cathodal HD-tsDCS and HD-tsACS. Both the right (F1,57 = 5.575; *p* = 0.022) and left (F1,57 = 6.802; *p* = 0.012) legs improved significantly after anodal HD-tsDCS compared with cathodal HD-tsDCS.

No significant correlations were found between intervention-induced changes in somatosensation and balance.

## 4. Discussion

This study investigated and compared the effects of three different HD-tsNICS protocols in supporting balance and deep as well as superficial sensitivities in healthy young adults. The data showed that (i) the spinal cord is an effective target for the application of NICS in supporting both balance and somatosensory control; (ii) the effects of HD-tsNICS differ significantly across protocols; (iii) despite the positive effects on all parameters, no significant correlations between balance and somatosensation improvement existed; and (iv) spinal NICS is a gentle and safe alternative to cranial NICS. In our study, both anodal HD-tsDCS and HD-tsACS supported balance and deep sensitivity. The supporting effect on superficial sensitivity of anodal HD-tsDCS compared with cathodal HD-tsDCS and HD-tsACS, but not with placebo HD-tsDCS/ACS, was demonstrated. Neither severe nor less severe adverse events were evoked by the protocols used. Controversy is a major part of trials investigating tDCS/tACS in human cohorts that reported less severe harmful effects such as itching, tingling, headache, discomfort, or burning sensation [50,51,52]. Overall, these results indicate that the spinal cord may be a gentle alternative to the brain for non-invasive neuromodulation techniques. Future studies should examine the long-term safety and tolerability of repetitive spinal cord modulation.

### 4.1. NICS in Supporting Balance

Numerous studies investigated the effectiveness of NICS in supporting balance control in healthy young [53,54,55,56,57,58,59,60,61,62,63,64,65,66,67] and elderly [68,69,70,71,72,73,74,75] adults up to now. However, existing experiments have shown limited variability regarding the targeted regions; further, the exact effects of NICS on the balance ability showed some ambiguity. Most studies involved the stimulation of either the cerebellum [53,54,55,56,59,62,63,66,67,68,69,70,71,75] or the M1 [57,58,60,64,67,69,70,72,73]. The supplementary motor area [65,74], dorsolateral prefrontal cortex [57,61], and dorsal premotor cortex [66] have rarely been examined. Other regions were not considered in existing trials. Even though a large number of the existing studies have demonstrated the supportive effects of NICS on balance control [53,55,57,60,64,67,68,69,70,71,72,74], an equally large number of trials failed to detect any significant effects [53,54,56,57,58,59,61,63,65,66,73,75]. Worsening of balance through NICS has only been detected in a small number of trials [54,62,65]. The existing body of evidence does not indicate the superiority of any region, as the absence of positive effects is evenly distributed across all investigated areas. This raises the question of whether other parts of the neural system are more effective targets for the application of NICS to support balance control. In a previous study, we compared the effects of NICS on the (i) cerebellum, (ii) M1, and (iii) the spinal cord [16]. Our data demonstrate that the spinal cord is a more effective target than both the cerebellum and M1 regions for the application of NICS in supporting balance control [16]. Thus, in the present study, we decided to examine the potential of spinal NICS more closely and to compare the effects of the three protocols. Overall, our data showed that both anodal DCS and ACS supported balance ability, whereas cathodal DCS did not induce any relevant effects. Despite criticism regarding the oversimplified theory that anodal DCS “supports” and cathodal DCS “suppresses” neural networks and abilities, our data did not completely fall out of this framework. A similar pattern was demonstrated in previous trials. In prior studies, a single session of anodal tDCS was found to induce either an improvement [55,57,60,64,67,68,69,70,71,72,74] or no effect [53,54,56,57,58,59,61,63,66,73,75], rarely worsening [62,65] balance. In contrast, a single session of cathodal tDCS worsened [54,62], had no effect [56,63,65], or scarcely improved [53] balance ability. The supportive effects of ACS on balance have not been previously investigated, and our results encourage a more intensive investigation of this neglected protocol. In addition, HD-NICS should be studied more frequently. Only a few studies have applied anodal HD-DCS to the cerebellum [66,67], dorsal premotor cortex [66], or M1 [67], with either positive [67] or no effects [66] on balance ability having previously been detected in healthy young adults. Our results encourage further investigations of this innovative protocol.

### 4.2. NICS in Modulation of Somatosensation

Only a few studies have investigated the modulatory effects of NICS on lower-limb sensitivity in both young [67,76,77,78,79] and elderly [80] healthy adults, as reported in a recent review [81]. Previous trials have reported inconsistent results regarding the methods, probands, and intervention-induced effects. Single [67,76,77,81] or multiple [78,79] sessions of anodal tDCS [67,77,78,79,80] or cathodal tDCS [75] have been applied over the M1 using either conventional [76,77,80] or HD [67,78,79] electrodes. Anodal tDCS has been shown to improve foot sole vibratory sensation [81] and passive ankle movement detection [78,79], while reducing pain pressure sensation over the rectus femoris muscle [77]. Some studies have not found any relevant effects of anodal tDCS on passive ankle movement detection [67], pressure sensation [77], or two-point discrimination [77] at the back of the foot [77]. Cathodal tDCS improves the pressure sensitivity of the distal pulp of the hallux [76]. This limited evidence calls for further research in this field. Overall, our data demonstrated that both anodal HD-tsDCS and HD-tsACS support the existence of vibration sensitivity compared to sham HD-tsDCS/ACS and cathodal HD-tsDCS. Anodal HD-tsDCS further improved pressure sensitivity compared to both cathodal HD-tsDCS and HD-tsACS, but not to sham HD-tsDCS/ACS. No significant effects were observed for cathodal HD-tsDCS compared with sham HD-tsDCS/ACS.

## 5. Conclusions

The present study focused on innovative and insufficiently investigated NICS methods, namely (i) spinal targeting, (ii) ACS in comparison to both cathodal DCS and anodal DCS, and (iii) HD DCS/ACS techniques. Our data shows that both anodal HD-tsDCS and HD-tsACS are beneficial in supporting balance and somatosensation, while cathodal HD-tsDCS did not induce beneficial effects. The generalisability of our findings is difficult due to the limited sample size (58 probands) as well as age (18–30 years) and health status (only healthy participants) restrictions. Future studies should verify the applicability of protocols investigated in our study on cohorts of different ages and health conditions. A closer look should be taken at interindividual differences (sex, weight, height, bone structure, subcutaneous fat tissue thickness), as well as several internal (e.g., circadian rhythm, hormones) and external (e.g., activities prior to or in parallel to stimulation) factors that may significantly influence the stimulation-induced effects. The application of neuroimaging and neurophysiological approaches, in addition to behavioural evaluations, may improve our understanding of brain–spinal cord interactions and support the development of targeted therapies in several cohorts. Stroke [7], Parkinson’s disease [82], multiple sclerosis [83], spinal cord injuries [84], and several other diseases are associated with H-reflex abnormalities in parallel to gait and/or balance disability. Non-invasive spinal modulation presents a promising approach for their rehabilitation.

## Figures and Tables

**Figure 1 biomedicines-12-02379-f001:**
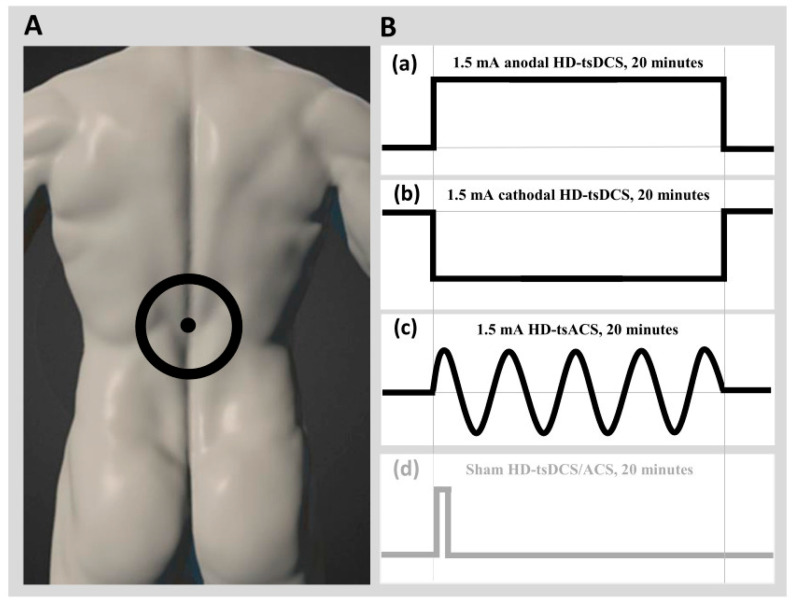
(**A**) Electrodes positioning and (**B**) stimulation protocols applied to each proband in a randomised order.

**Figure 2 biomedicines-12-02379-f002:**
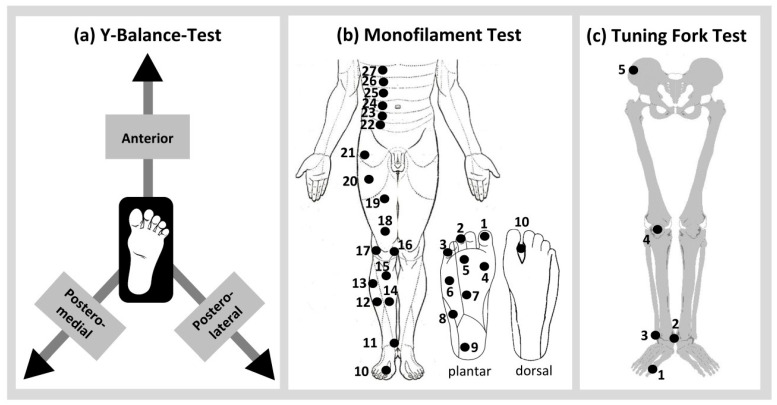
The (**a**) Y Balance Test, (**b**) Monofilament Test, and (**c**) Tuning Fork Test were performed immediately prior to and after each intervention.

**Figure 3 biomedicines-12-02379-f003:**
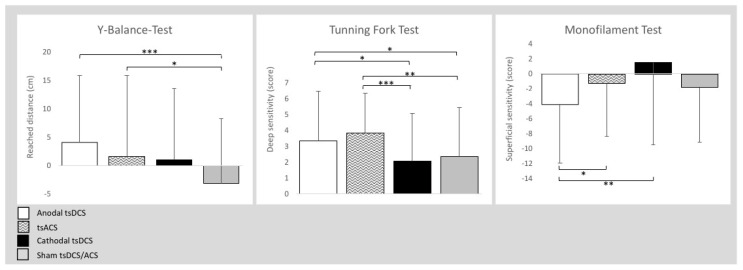
Intervention-induced changes (means and SD) in the Y Balance Test, Tuning Fork Test, and Monofilament Test in relation to the baseline. Notes: cm = centimetre; * ≤ 0.05; ** = *p* ≤ 0.01; *** = *p* ≤ 0.001.

**Table 1 biomedicines-12-02379-t001:** Balance and sensitivity (means and SD) at pre- and post-interventions.

			Anodal HD-tsDCS	HD-tsACS	Cathodal HD-tsDCS	Sham HD-tsDCS/tsACS
Y Balance Test/reach distance (cm)	total	pre	525 ± 63	524 ± 57	521 ± 60	528 ± 65
		post	530 ± 66 ***^S^	526 ± 60 *^S^	522 ± 61	525 ± 65
	right leg	pre	263 ± 32	262 ± 30	261 ± 31	264 ± 33
		post	266 ± 34 *^S^	263 ± 31	261 ± 31	262 ± 33
	left leg	pre	263 ± 33	262 ± 28	261 ± 30	264 ± 33
		post	265 ± 33 ***^S^	263 ± 29	262 ± 29	263 ± 32
Tuning Fork Test/score	total	pre	48.24 ± 9.24	47.33 ± 8.90	47.74 ± 11.40	48.16 ± 10.20
		post	51.59 ± 10.36 *^S,^*^C^	51.16 ± 9.53 **^S,^***^C^	49.02 ± 10.89	50.52 ± 10.84
	right leg	pre	23.97 ± 4.75	23.96 ± 4.74	23.67 ± 24.70	23.92 ± 5.37
		post	25.63 ± 5.26	25.74 ± 4.81 *^C^	24.40 ± 5.38	25.14 ± 5.55
	left leg	pre	24.29 ± 4.46	23. 63 ± 4.64	24.07 ± 5.81	24.24 ± 5.05
		post	25.97 ± 5.28 *^S^	25.57 ± 4.97 *^S,^*^C^	25.20 ± 5.70	25.37 ± 5.49
Monofilament Test/score	total	pre	156 ±17	156 ± 17	151 ± 15	155 ± 18
		post	152 ± 19 **^C,^*^Al^	154 ± 19	152 ± 17	153 ± 18
	right leg	pre	78.8 ± 8.7	78.0 ± 8.5	76.3 ± 7.6	78.2 ± 9.1
		post	76.6 ± 9.2 *^C^	77.5 ± 9.3	76.9 ± 9.0	77.3 ± 8.8
	left leg	pre	77.8 ± 9.1	78.0 ± 9.3	75.0 ± 7.4	76.9 ± 9.1
		post	75.8 ± 9.3 *^C^	77.2 ± 9.9	76.0 ± 8.3	76.1 ± 9.4

Notes: mm^2^ = square millimetre; ms = millisecond; S = significant intervention-induced changes compared to sham HD-tsDCS/ACS; C = significant intervention-induced changes compared to cathodal HD-tsDCS; Al = significant intervention-induced changes compared to HD-tsACS; * *p* ≤ 0.05; ** *p* ≤ 0.01; *** *p* ≤ 0.001.

## Data Availability

The datasets generated and/or analysed in the current study are available from the corresponding author upon reasonable request.

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
