# Peer review of "High-Definition Trans-Spinal Current Stimulation Improves Balance and Somatosensory Control: A Randomised, Placebo-Controlled Trial"

_biomedicines, 2024, doi:10.3390/biomedicines12102379_

Round 1

Reviewer 1 Report

Comments and Suggestions for Authors

The paper presents the results of using three different HD-NICS protocols delivered to the spinal cord. Overall, the paper is well-written and the findings are valuable in the domain of NICS for improving balance and somatosensory abilities. I have only a few minor suggestions that could improve the presentation of the methods and results.

Major comments:

1.      For section 2.3 Intervention, it would be beneficial if the authors add a picture of the setup. This would help clarify the electrode layout and facilitate replication by other researchers.

2.      Figure 1A. Please add time axis below each subfigure, as it is currently unclear how long each stimulation lasted.

3.      Table 1 should be improved by adding gridlines to enhance readability.

Author Response

Dear reviewer,                                                                                                           

thank you for the time taken to review our manuscript.

We have revised the manuscript in accordance with them and hope that our manuscript meets your requirements.

Comment 1: For section 2.3 Intervention, it would be beneficial if the authors add a picture of the setup. This would help clarify the electrode layout and facilitate replication by other researchers.

Response 1: Thank you for this hint. We included a figure demonstrating electrodes positioning in the Figure 1.

Comment 2: Figure 1A. Please add time axis below each subfigure, as it is currently unclear how long each stimulation lasted.

Response 2: The figure was revised in accordance with the comment.

Comment 3: Table 1 should be improved by adding gridlines to enhance readability.

Response 3: The Table was revised accordingly.

Reviewer 2 Report

Comments and Suggestions for Authors

While reviewing the manuscript “High-Definition Trans-Spinal Current Stimulation Improves Balance and Somatosensory Control: A Randomised, Placebo-Controlled Trial” This study reveals the potential of these techniques to improve balance and somatosensory performance in healthy young adults by comparing different types of spinal cord current stimulation, providing an important scientific basis for future applications in patients, however, I would like to mention several issues, and I think it should be solved before I suggest publishing this manuscript.

Comment 1: The study sample size was 58 students, although sufficient for a pilot study, the sample size was relatively small and the subjects were all healthy young adults. As such, generalizability of the findings may be limited. It is recommended to expand the sample size in future studies and consider including subjects of different ages and health conditions to verify the broad applicability of these findings.

Comment 2: Are individual differences (e.g. sex, weight, height, baseline level, etc.) likely to have an impact on the results? Have subgroup analyses been considered to investigate the impact of these factors on the results?

Comment 3: It is mentioned that the spinal cord is involved in balance and postural control through the inhibition of H-reflex. Could it be further explained how inhibition of the H-reflex specifically impacts balance ability, whether this inhibition is long-term adaptation or short-term regulation?

Comment 4: There were no serious or minor adverse events in the study. Does this mean that these stimulus packages are also safe in the long run? Are long-term follow-up studies planned to assess long-term safety?

Comment 5: Is it planned to further investigate the regulatory mechanisms of these stimulation protocols on spinal cord and brain networks in combination with neuroimaging or neurophysiological approaches?

Comment 6: The results of the study showed the potential benefit of HD-NICS in healthy people, but its effect in clinical patients is not clear. It is recommended to explore the effect of these stimulation protocols in patients with different diseases, such as spinal cord injury, stroke and multiple sclerosis, in future studies to assess their potential in rehabilitation.

Comments on the Quality of English Language

 Minor editing of English language required.

Author Response

Dear reviewer,

thank you for the time taken to review our manuscript.

We have revised the manuscript in accordance with them and hope that our manuscript meets your requirements.

Comment 1: The study sample size was 58 students, although sufficient for a pilot study, the sample size was relatively small, and the subjects were all healthy young adults. As such, generalizability of the findings may be limited. It is recommended to expand the sample size in future studies and consider including subjects of different ages and health conditions to verify the broad applicability of these findings.

Response 1: Thank you for this hint. We fully agree with them. The Conclusions chapter was revised in accordance.

Line 295: The generalizability of our findings difficult, due to the limited sample size (58 probands) as well as age (18–30 years) and health status (only healthy participants) restrictions. Future studies should verify the applicability of protocols investigated in our study on cohorts of different ages and health conditions.

Comment 2: Are individual differences (e.g. sex, weight, height, baseline level, etc.) likely to have an impact on the results? Have subgroup analyses been considered to investigate the impact of these factors on the results?

Response 2: In fact, the factors mentioned may have impact the intervention-induced effects, but our study is a pivotal project with limited number of probands and no subgroup analyses have been considered. We included a paragraph in the Conclusions chapter to this topic.

 Line 298: A closer look should be taken at interindividual differences (sex, weight, height, bone structure, subcutaneous fat tissue thickness), as well as several internal (e.g., circadian rhythm, hormones) and external (e.g., activities prior to or in parallel to stimulation) factors that may significantly influence the stimulation-induced effects.

Comment 3: It is mentioned that the spinal cord is involved in balance and postural control through the inhibition of H-reflex. Could it be further explained how inhibition of the H-reflex specifically impacts balance ability, whether this inhibition is long-term adaptation or short-term regulation?

Response 3: Thank you for this hint. We revised the Introduction chapter in accordance.

Line 65: In fact, numerous studies demonstrated that both balance training and improved balance control are accompanied by suppressed H-reflex. In contrast, disability-induced worsening of balance control correlates with H-reflex increase. E.g., a single training session on a balance platform induced, in addition to an improved balance ability, a short-term suppression of the H-reflex in healthy elderly [5]. Similarly, balancing on a narrow beam induced, as compared to normal walking, a short-term reduction of the H-reflex amplitude in healthy people [6]. Stroke patients demonstrated, in parallel with gait and balance disability, a long-term increase of H-Reflex amplitude compared to healthy peers [7]. The successful balance rehabilitation in a stroke cohort was associated with H-reflex normalization [8]. It is cogitable that the reactions of the spinal reflex system during postural disturbances are not always adequate. The reflex-mediated excessive stabilizing movements often paradoxically lead to a destabilization. A suppression of the spinal reflexes may increase the involvement of supraspinal centres during motor control and prevent reflex-mediated destabilizing movements [3].

Comment 4: There were no serious or minor adverse events in the study. Does this mean that these stimulus packages are also safe in the long run? Are long-term follow-up studies planned to assess long-term safety?

Response 4: We included a sentence in the discussion chapter.

Line 235: Future studies should examine the long-term safety and tolerability of repetitive spinal cord modulation.

Comment 5: Is it planned to further investigate the regulatory mechanisms of these stimulation protocols on spinal cord and brain networks in combination with neuroimaging or neurophysiological approaches?

Response 5: The information was included.

Line 302: The application of neuroimaging and neurophysiological approaches, in addition to behavioral evaluations, may improve our understanding of brain-spinal cord interactions and support the development of targeted therapies in several cohorts.

Comment 6: The results of the study showed the potential benefit of HD-NICS in healthy people, but its effect in clinical patients is not clear. It is recommended to explore the effect of these stimulation protocols in patients with different diseases, such as spinal cord injury, stroke and multiple sclerosis, in future studies to assess their potential in rehabilitation.

Response 6: The information was included.

Line 304: Stroke [7], Parkinson's disease [84], multiple sclerosis [85], spinal cord injuries [86] and several other diseases are associated with H-reflex abnormalities in parallel to gait and/or balance disability. Non-invasive spinal modulation presents a promising approach for their rehabilitation.

Line 552:

  1. Sabbahi M, Etnyre B, Al-Jawayed IA, Hasson S, Jankovic J. Methods of H-reflex evaluation in the early stages of Parkinson's disease. J Clin Neurophysiol. 2002; 19: 67-72.
  2. Cantrell GS, Lantis DJ, Bemben MG, Black CD, Larson DJ, Pardo G, Fjeldstad-Pardo C, Larson RD. Relationship between soleus H-reflex asymmetry and postural control in multiple sclerosis. Disabil Rehabil. 2022; 44: 542-548.
  3. Thompson AK, Wolpaw JR. H-reflex conditioning during locomotion in people with spinal cord injury. J Physiol. 2021; 599: 2453-2469.

Round 2

Reviewer 2 Report

Comments and Suggestions for Authors

Authors have adequately addressed all review comments and made corresponding improvements in the revised manuscript. Based on this, I agree that this manuscript can be published and no further suggestions for revision will be made.